# Chemical Characterization and Metagenomic Identification of Endophytic Microbiome from South African Sunflower (*Helianthus annus*) Seeds

**DOI:** 10.3390/microorganisms11040988

**Published:** 2023-04-10

**Authors:** Fatai Oladunni Balogun, Rukayat Abiola Abdulsalam, Abidemi Oluranti Ojo, Errol Cason, Saheed Sabiu

**Affiliations:** 1Department of Biotechnology and Food Science, Durban University of Technology, Durban 4000, South Africa; balogunfo@yahoo.co.uk (F.O.B.); abiola4rusul@gmail.com (R.A.A.); 2Centre for Applied Food Sustainability and Biotechnology, Central University of Technology, Bloemfontein 9300, South Africa; 3Department of Animal Science, University of the Free State, Bloemfontein 9300, South Africa; casoned@ufs.ac.za

**Keywords:** endophyte, bacteria diversity, fungi diversity, microbiome, next-generation sequencing, sunflower seeds

## Abstract

*Helianthus annus* (sunflower) is a globally important oilseed crop whose survival is threatened by various pathogenic diseases. Agrochemical products are used to eradicate these diseases; however, due to their unfriendly environmental consequences, characterizing microorganisms for exploration as biocontrol agents are considered better alternatives against the use of synthetic chemicals. The study assessed the oil contents of 20 sunflower seed cultivars using FAMEs-chromatography and characterized the endophytic fungi and bacteria microbiome using Illumina sequencing of fungi ITS 1 and bacteria 16S (V3–V4) regions of the rRNA operon. The oil contents ranged between 41–52.8%, and 23 fatty acid components (in varied amounts) were found in all the cultivars, with linoleic (53%) and oleic (28%) acids as the most abundant. Ascomycota (fungi) and Proteobacteria (bacteria) dominated the cultivars at the phyla level, while *Alternaria* and *Bacillus* at the genus level in varying abundance. AGSUN 5102 and AGSUN 5101 (AGSUN 5270 for bacteria) had the highest fungi diversity structure, which may have been contributed by the high relative abundance of linoleic acid in the fatty acid components. Dominant fungi genera such as *Alternaria*, *Aspergillus*, *Aureobasidium*, *Alternariaste*, *Cladosporium*, *Penicillium*, and bacteria including *Bacillus, Staphylococcus*, and *Lactobacillus* are established, providing insight into the fungi and bacteria community structures from the seeds of South Africa sunflower.

## 1. Introduction

Microbiomes are plant-associated microorganisms (including fungi, bacteria, viruses, and algae) [1]. While some of these microorganisms are found on the surface of the plant (epiphytes), the majority of them, referred to as endophytes, exist inside plant tissues and fungi organisms forming the largest microbial structure of these communities [2]. Endophytes, ubiquitous in nature, have the ability to colonize every part of plant including the seeds and are endowed with various metabolites (phenolics, alkaloids, saponins, terpenoids, and steroids) habouring arrays of bioactive compounds of medicinal and pharmacological importance [3]. The mutual interaction between these endophytic organisms and the host plants has been established [2,4,5]. Apart from endophytes’ involvement in atmospheric nitrogen fixation, making host plants resistant to biotic and abiotic factors and producing plant hormones, they also contribute significantly to plant growth and development [6]. Additionally, they assist in removing soil contaminants and aid in the tolerability of soils with little or low fertility [5]. Specifically, endophytic fungi are endosymbionts to plants [7] and use this relationship to facilitate plants’ growth and development [8]. Despite the enormous benefits of an endophytic mycobiome, few others, such as *Fusarium chlamydosporum*, *Fusarium solani*, *Plasmopara halstedii*, *Puccinia helianthi*, *Phosmosis helianthi*, *Macrophonuria phaseolina*, *Verliallium dahilae*, and *Sclerotnia schetiorum* may be pathogenic [9]. Seeds are one of the important vegetative parts of a plant and harbour a high microbial load [10]. The beneficial microbes prevent seed degradation [10] and enhance their germination or survival rate, root length, and biomass deposition [11]. However, crop seeds can be infected; the sources of infection are usually from the environment (soil, air, and storage places) [12]. The success of any crop production activity lies in crop seeds planted for eventual germination. While seeds that are pathogen-free are essential, those that are prone to diseases bring about weakened plant growth resulting in reduced productivity, poor crop yield, and huge financial implications in combating the disease [13]. Hence, the adequate treatment of these seeds before immersing them in the soil is vital to avert inadequacies in crop yield and financial losses. The pre-treatment of these seeds with various chemicals, including fungicides and pesticides (Vitavax, Stiletto, Thiram), has side effects as chemicals are not eco-friendly due to the possibility of leaching into underground water bodies and becoming threats to aquatic organisms, killing important soil organisms, the possible contamination of crop products arising from the accumulation of these chemicals and the likelihood of crop pest resistance. Hence, searching for an eco-friendly alternative to treat seeds in order to enhance agricultural or food sustainability is vital [14,15]. Interestingly, in recent times, biological control involving the use of microorganisms is now explored as an eco-friendly alternative to the use of agrochemical products [16]. Therefore, isolating and characterizing the microbiome in seeds such as *Helianthus annus* seeds should provide insights into a possible exploration into being formulated as a biocontrol agent that is eco-friendly and against the use of synthetic chemicals in getting rid of these pathogenic organisms.

*Helianthus annus* (commonly called sunflower) is regarded as an essential (oilseed) crop with global-scale cultivation and importance [17]. In fact, it is the third most important oilseed, following soybeans and rapeseed and the fourth most essential vegetable oil-containing crop worldwide (after palm, soybean, and rapeseed) [18], with global production pegged at 56.97 million tonnes in 2021 arising from 28.27 million hectares of land [18]. While this oilseed crop is largely cultivated in Ukraine, Russia, Turkey, Argentina, the United States, South Africa etc., the annual production in South Africa is between 600,000 to 800,000 tonnes, contributed largely from four Provinces (North-West, Free State, Limpopo, and Mpumalanga) [19]. The seeds have many nutritional benefits; they are sometimes blended for several purposes (bread, snacks, etc.) or combined with vegetables to make salad garnish [20]. The seeds are extracted to produce oil containing protein (20%) and fat (36–50%) [21]. Though there may, however, be variation in the amount of the fats in oils depending on species type or cultivars, ninety percent of these fat components are unsaturated, consisting of oleic (16–19%) and linoleic (68–72%); the remaining 10% are saturated fats comprising palmitic (6%), stearic (4%) including myristic, myristoleic, palmitoleic, arachidic, and behenic acids [22].

Genetic disposition and environmental factors contribute to the variation in the oil or fat contents of sunflower varieties [22,23,24], and these factors also influence the diversity of the cultivars [17]. Despite the huge economic benefits of this important oilseed crop, it has only received little attention on diseases and contamination associated with the plant or its seeds from South Africa caused by arrays of pathogenic, endophytic fungi and parasite species [25,26,27]. Assessments using next-generation sequencing (NGS) provide in-depth information about the microbial community that could be beneficial in controlling the seed pathogen [28]. Adeleke et al. [29] assessed the bacterial community structure on the root and stem of a South African sunflower cultivar (PAN 7160 CLP) using the NGS approach, but the identification of the endophytic mycobiome in the plant seed using a deep amplicon sequencing approach has not been done.

Interestingly, to the best of our understanding, the fungi and bacteria diversity of South African sunflower seeds using the deep amplicon method has not been employed. Thus, the present study aimed to characterize the endophytic microbiome structure in 20 South Africa sunflower seeds. Hence, in order to provide comprehensive information on the microbiome structure of South African sunflower seeds, firstly, the oil contents and fatty acids components of 20 cultivars of sunflower seeds will be determined in FAME-chromatography techniques. Secondly, the fungi and bacteria community compositions and structure will be assessed using the Illumina MiSeq sequencing platform. Thirdly the possible influences or roles these fatty acids components play in determining microbial community structures of coatless seeds will be established.

## 2. Materials and Methods

### 2.1. Materials

#### Seeds Collection

Seed samples, pre-treated with fungicides (to prevent seed-borne pathogens) from 20 cultivars of sunflower, were sourced from the Agricultural Research Council Grain Crop Institute (Potchefstroom, South Africa) designed by different manufacturers. The identification of the seeds are as follows; AGSUN 5108 CLP, AGSUN 5270, AGSUN 5102, AGSUN 8251, AGSUN 5106 CLP, AGSUN 5101 CLP, AGSUN 5103 CLP (from Agricol); P65LP54, P65LL02, P65LP65 (Pioneer); PAN7160 CLP, PAN7102 CLP, PAN7170, PAN7158 HO, PAN7180 CLP, PAN7156 CLP (Pannar); LG5710, LG5678 CLP, AGUARA (LimaGrain); and SY3970 (Sensako).

### 2.2. Methodology

#### 2.2.1. Oil Content Determination

The determination of the oil contents of the cultivars was initiated with the preparation of the seeds, followed by Soxhlet extraction, and finally, fatty acids methyl esters (FAMEs)-chromatographical analyses.

##### Seed Preparation

Adopting the method of Larson [30] with much modification, 5 g of the seeds were weighed, soaked in distilled water for 1 min, and then pealed to remove the seed coat. Following this, the seeds were rinsed in distilled water to remove potential dirt obtained during the seedcoat removal and then dried at 37 °C for 24 h. The dried seeds were then lightly grounded in mortar and pestle to increase the surface area before being subjected to Soxhlet extraction.

##### Soxhlet Extraction

The seeds were extracted as earlier reported [31,32] with slight modifications. The grounded seeds of each cultivar were placed in the thimble before suspending in the Soxhlet extractor. Petroleum ether (250 mL, 40/60 strength) was measured and introduced into the round bottom flask suspended on a heater for (55 °C) 4 h. The petroleum ether-containing oil was concentrated on a rotary evaporator and water bath at 45 °C to remove or recover the petroleum ether to obtain the oil, which was collected and weighed to determine the individual oil contents of the cultivars.

##### Fatty Acids Methyl Esters (FAMEs)

The method described by McCurry [33] was used for FAMEs analysis. Briefly, 2 mL of chloroform: methanol (2:1) was added to ca. 100 mg oil. The mixture was vortexed and sonicated at room temperature for 30 min and thereafter centrifuged at 3000 rpm for 1 min. Approximately 130 µL of the bottom layer (chloroform) was dried completely with a gentle stream of nitrogen, reconstituted and vortexed with 100 µL of methyl tert-butyl ether (MTBE) and 30 µL of trimethyl sulfonium hydroxide (TMSH). Then, 1 µL of the derivatized mixture was injected in a 5:1 split ratio onto the gas chromatography (6890 N, Agilent Technologies)-flame ionization detector equipment (GC-FID) for further chromatographic separation.

##### Chromatographic Identification

Separation of the FAMEs was performed on a polar capillary column (RT-2560) with specifications (100 m, 0.25 mm ID, 0.20 µm film thickness) (Restek, Bellefonte, PA, USA). Helium was adopted as the carrier gas at a flow rate of 1 mL/min. The injector temperature was maintained at 240 °C. The oven temperature was programmed to 60 °C for 1 min, ramped to 120 °C at a rate of 8 °C/min for 1 min, followed by a ramping rate of 1.5 °C/min to 245 °C for 1 min, and a final ramped up to 250 °C at a rate of 20 °C/min for 2 min. Based on this, the various fatty acid constituents and compositions in each cultivar were determined by their retention times or elution (from the column based on the respective carbon numbers or atoms) measurement through the detector [33].

#### 2.2.2. Microbial Characterization

##### Surface Sterilization

The surface sterilization of the seeds was carried out based on the method of Kinge et al. [12] with slight modifications. Five grams of the various seeds were surface sterilized in 3% sodium hypochlorate for 3 min, this was followed by rinsing in sterile distilled water for 1 min, and thereafter the seeds were exposed to 2 min 70% ethanol and rinsed again for 1 min. The seed coats were then removed and subjected to 1 min each of 70% ethanol exposure and distilled water cleaning to get rid of the possible exposed microorganisms during the seedcoat removal, subsequently freeze-dried under controlled conditions and powdered under liquid nitrogen before DNA extraction.

##### DNA Extraction, PCR Amplification and Sequencing

The extraction of the gDNA was achieved based on the procedure described in the Nucleospin DNA PowerSoil extraction kit (Mo BIO Laboratories, Inc., Carlsbad, CA, USA). Following this, DNA concentration and quality were determined using a NanoDrop fluorospectrometer (Thermo Scientific, Johannesburg, GP, South Africa) and 0.8% Agarose gel electrophoresis, respectively. The gDNA obtained was amplified on an internal transcribed spacer (ITS1) of the rRNA gene using forward (CTTGGTCATTTAGAGGAAGTAA) and reverse (GCTGCGTTCTTCATCGATGC) primers for fungi while for the bacteria, the amplification was on the V3-V4 region of the 16S rRNA ribosomal gene using 319 forward (CCTACGGGNGGCWGCAG) and 806 reverse (GACTACHVGGGTATCTAATCC) primers designed (Inqaba Biotechnological Industries, Pretoria, GP, South Africa) according to Usyk et al. [34]. The DNA amplification procedure was achieved through Eyre et al. [35] using G-Storm thermal cycler (Somerton Biotechnology, Bristol, UK). The choice of the ITS 1 region of the rRNA gene over ITS 2 was due to its variability and/or longer reads length and more GC content [36], as well as its wider acceptance in fungi metagenomic studies. The PCR products (fungi and bacteria) were then used to prepare a DNA library following the Illumina TruSeq^®^ Nano DNA Sample Preparation protocol (version 2) for subsequent sequencing, which was achieved on an Illumina MiSeq platform, using a 2 × 301 cycle paired-end approach at the Molecular Research LP (MR DNA) lab, Shallowater, TX, USA.

##### Data Processing and Statistical Evaluation

The raw sequencing reads obtained were analyzed using QIIME 2 (https://qiime2.org (accessed on 20 September 2022)) [37]. Paired-end sequences imported into QIIME 2 were quality-controlled and combined using DADA2. This grouped sequences into amplicon sequence variants (ASVs) based on 99% sequence similarity. After building the ASV table and removing chimeras, taxonomy for the 16S and ITS reads were assigned using the SILVA 132 [38] and UNITE [39] databases, respectively. The alpha rarefaction curve was plotted with 1000 sampling depths.

The processed gene sequence data was further analyzed using R-Studio, an integrated development environment (IDE) for the programming language R [40]. Phyloseq [41], Tidyverse [42], Vegan [43], and Biomformat [44] were used during the processing of the taxonomic data. For beta diversity analysis, the taxonomic structures of the microbial communities were visualized using principal coordinate analysis (PCoA). To assess beta diversity differences between different seed providers, a permutational analysis of variance (PERMANOVA) was used. These analyses were performed with the “adonis” functions in vegan for R. A principal component analysis (PCA) plot was explored using the Bray–Curtis distance matrix in order to determine the distribution of the fatty acid components. The design of the PCA plot was achieved with version 5 of CANOCO software (Microcomputer Power, Ithaca, NY, USA). One-way analysis of variance (ANOVA), supported by Bonferroni’s multiple comparison test, were used for the analysis of the chemical components of the oils using Graph Pad Prism version 9.5.1 for Windows software, Graph Pad software, San Diego, CA, USA. Results are presented as mean ± standard error of mean (SEM) of triplicate determination. Statistical significance was considered at (*p* < 0.05).

## 3. Results

### 3.1. Fats Composition

The oil content of the cultivars ranged from 41 to 52.8% (Table 1). A total number of 23 fatty acids identified in all the cultivars comprised 14 saturated (such as caproic, caprylic, capric, lauric, myristic, pentadecyclic, palmitic, margaric, stearic, arachidic, heneicosylic, behenic, tricosylic, and lignoceric acids) and nine unsaturated (oleic, linoleic, paullinic, dihomo-a-linoleic, eicosatetraenoic, eicosapentaenoic, nervonic acids) fatty acids (Figure 1). Linoleic and oleic acids were the most abundant (53.12 and 28.42%, respectively) in all the cultivars, followed by palmitic and stearic acids and other fatty acids in lesser or negligible amounts (Figure 1, Appendix A) as reflected in the average mean values (Appendix A). Cultivar-wise, PAN7180 CLP revealed the most abundance and least abundance of linoleic (69.84%) and oleic (15.44%) acids, respectively, while the least abundance of linoleic acid was found with PAN7158 HO (4.97%) and having 86.03% in oleic acids, reflecting its highest abundance (Figure 1). The principal component analysis (PCA) of oil contents and fatty acids components revealed two distinct clusters containing representatives from all the samples (Figure 2). Clustered samples contain similar oil contents and fatty acid compositions. However, the Aguara sample from LimaGrain is a clear outlier, more than likely due to the samples’ higher percentage of caproic, caprylic, and capric acids compared to the rest of the samples. The plot revealed a variation of 74.2% between the x and y-axes (51.6% and 22.6%, respectively).

### 3.2. Fungal Diversity and Abundance

A total of 227 fungal (ASVs), ranging from 16 to 40 ASVs per sample, were found using identical sequencing depth in all samples. Rarefaction curves, Chao1, and Good’s coverage estimates suggest that this sequencing depth was adequate to capture most of the diversity in each sample (Appendix A). Of the total number of ASVs, no ASVs were unique to Agricol and Pioneer samples, 28 ASVs (representing 0.32% of the total number of sequences) were unique to LimaGrain, 38 ASVs (0.60%) were unique to Pannar and 5 ASVs to Sensako while 14 (94.02%) ASVs were shared among all the sources of the seeds (Figure 3). Additionally, the number of ASVs were highest in Agricol and Pioneer cultivars (109) indicating their highest richness followed by Pannar (82), LimaGrain (63) and the lowest richness (27) attributed to Sensako (Figure 3).

At the phylum level, Ascomycota was the most abundant (41.9%) in all samples (Figure 4, Appendix A), with the highest in AGSUN 5270 (90.63%) and the lowest in AGSUN 5106 CLP (6.15%); this was followed by Basidiomycota (1.42%) and Mucoromycota (0.18%) while a larger percentage are unidentified (56.5%) (Appendix A). Specifically, the relative abundance of Ascomycota in AGSUN 5270, P65LP54, and AGUARA were 90.63%, 85.56%, and 80.93%, respectively. Basidiomycota found mostly in the Agricol cultivars depicted moderate abundance (>1% frequency) with AGSUN 5101, revealing the highest diversity (7.83%) and the least (diversity) or absence in AGUARA (0%). Additionally, at >1% frequency, Mucoromycota was only seen in AGSUN 8251 (1.29%) (Appendix A). Ascomycota showed seven fungi classes (Sordariomycetes, Dothideomycetes, Leotiomycetes, Saccharomycetes, Eurotiomycetes, Arthoniomycetes, Pezizomycotina_cls_Incertae_sedis) with Dothideomycetes being most abundant and 21 orders. Basidiomycota was found to have six (Ustilaginomycetes, Agaricomycetes, Cystobasidiomycetes, Malasseziomycetes, Tremellomycetes, Microbotryomycetes) classes, having Agaricomycetes with the most occurrence and 14 order fungi communities (Appendix A).

Ascomycota and Basiodiomycota revealed diverse families (30 and 16, respectively) and genera (43 and 16, respectively) (Appendix A). The most dominant fungal member at the genus level was unidentified (56.83%) based on the UNITE database; however, a BLASTn search of the representative sequence indicated that it is the ITS sequence for the sunflower (*Helianthus* sp.). Thus, the most dominant fungal genus in all samples is *Alternaria* spp. (Figure 5, Appendix A), accounting for a relative abundance of 30.7%, followed by *Aureobasidium* (2.08%), *Aspergillus* (0.68%), and *Alternariaste* (0.53%) among the identified genera present. Interestingly, cultivars (AGSUN 5270, P65LP54, and AGUARA) with the highest abundance at the phyla level exhibited a similar trend at the genus level with a relative abundance of 89.44%, 77.45%, and 75.34%, respectively, while the least abundance was found with the pioneer cultivars [P65LL02 (0.96%) and P65LP65 (1.14%)], as well as AGSUN 8251 (3.24%). Overall, a relatively higher abundance of the genus *Alternaria* was observed with Pannar and the Agricol cultivars. The genera were observed to exhibit some level of colonization (determined by lack/low/high abundance) in the samples. Typically, *Aureobasidium* was abundant in PAN7158 (24.19%), AGSUN 5102 CLP (9.84%), and SY3790 (3.94%) while low (<1%) in the remaining cultivars. *Cladosporium* was absent (0%) in some Agricol cultivars (AGSUN 5102 CLP, AGSUN 5108 CLP and AGSUN 5270) and low (<1%) in Pannar, Pioneer, and Sensako cultivars. *Eremothecium* abundance was very high in Pioneer P65LL02 (51.23%) cultivar, while *Alternariaster* was also observed to reflect moderate abundance (7.1%) in AGSUN5108 CLP; however, the relative abundance of these two genera was very low (<1%) in remaining cultivars.

The results of diversity analyses showed no difference in alpha diversity metric as observed between the seed companies (Figure 6). However, the result cultivar-wise showed the highest fungi richness in the Agricol cultivars (Figure 6 Left). The Shannon alpha diversity metric saw the seed companies revealing the highest diversity and with about 17 of the cultivars depicting a Shannon index above 1.2 (Figure 6 Middle) as well as a Simpson index above 0.5 (Figure 6 Right). Additionally, while the cultivars seem to be scattered into four positions on the plot (Figure 7), the type of cultivar has no significant effect on the structural differences among fungal communities. No distinct fungal communities (ASV level) were detected between the seed companies (PERMANOVA F_4,19_ = 0.78, R^2^ = 17%, and *p* = 0.683).

### 3.3. Bacteria Diversity and Abundance

Using the same sequencing depth in all the samples, a total number of 103 bacterial ASVs, ranging from 2 to 20 ASVs per cultivar, were obtained. The sequencing depth was enough to accommodate in each sample most of the diversity based on the rarefaction curves, Chao1 and Good’s coverage estimates (Appendix A). Of the total number of ASVs, no ASVs were unique to Agricol and Pioneer samples; however, 3 ASVs (representing 0.40% of the total number of sequences) were unique to LimaGrain and 19 ASVs (4.77%) were unique to Pannar, and 1 ASV attributed to Sensako while no ASV was shared between all the seed companies (Figure 8). In a similar trend to fungi richness arising from the cultivars, Agricol and Pioneer depicted the highest richness, having 46 ASVs each followed by Pannar (22), LimaGrain (6) and lastly Sensako with only 1 ASV.

Proteobacteria was found in all the cultivars with a relative abundance of 58.68%. It was observed that the genus completely dominated (100%) most of the Pannar (PAN7160 CLP, PAN 7158 HO, PAN 7156 CLP, PAN 7180 CLP) and Pioneer (P65LP65, P65LP54), and LimaGrain (AGUARA, LG5678 CLP) cultivars at the Phylum level. Proteobacteria is followed by Firmicutes and is seen to exhibit higher abundance in most of the Agricol cultivars (15.57%) compared to other brands [Pannar (3.15%), Pioneer (4.86%)] (Figure 9, Appendix A). Gammaproteobacteria was the only attributed bacteria community at the class level for Proteobacteria, whereas Firmicutes was found to be endowed with three (Bacilli, Erysipelotrichia and Clostridia) bacteria communities. However, at the order level, Bacillales, Lactobacillales, Erysipelotrichales, and Clostridiales were found attributed to Firmicutes with the former two bacteria as members of Firmicutes. Additionally, only Betaproteobacteriales and Pseudomonadales belongs to phylum Proteobacteria at the order level (Appendix A). However, Proteobacteria did not reflect diverse bacteria families, as Neisericeae and Moraxellaceae are the only two groups assigned to it; however, Firmicutes was endowed with diverse groups of nine families (Bacillaceae, Streptococcaceae, Lactobacillaceae, Staphylocaceae, Peptococcaceae, Paenibacillaceae, Erysipelotrychiaceae, Clostridiaceae, and a group assigned as Family_XI). At the genus level, 19 genera were identified from the 20 cultivars with *Bacillus* being dominant and abundant in most Agricol seeds cultivars (Figure 10). Four Agricol cultivars [AGSUN 5102 CLP (100%), AGSUN 8251 (100%), AGSUN 5101 (87.8%), and AGSUN 5106 (64.87%)] reflected good dominance of *Bacillus*. Other prominent genera are *Staphylococcus*, *Lactobacillus*, and *Acinetobacter*. Typically, Pannar cultivars (PAN 7158 and PAN 7170) were observed to be completely dominated by *Corynebacterium*_1 and *Paenibacillus*, respectively, while Agricol cultivars [AGSUN 5103 (100%), AGSUN 5108 (55.56%, 44.44%)] revealed a higher abundance of *Staphylococcus*, *Erysipelotrichaceae*_UCG-003, and *Acinetobacter*, respectively (Appendix A).

As observed with fungi diversity analyses, there was no difference in the alpha diversity metric of bacteria as noted between seed companies (Figure 11), though the highest richness (Figure 11 Left) and diversity (Figure 11 Middle and Right) of Agricol seeds are noticeable. The principal coordinate analysis revealed the clustering of the samples into three distinct positions (Figure 12). However, there was no significant effect in explaining structural differences among bacteria communities, and no evidence or existence of distinct bacterial communities (ASV level) between the seed companies (PERMANOVA F_4,19_ = 1.01, R^2^ = 22%, and *p* = 0.459) were established.

## 4. Discussion

Sunflower oil is endowed with a laudable nutrition effect in addition to its therapeutic advantage and economic importance [17]. The Soxhlet extraction of the seeds yielded variation in the oil contents among the cultivars, and the observed differences in this study could be due to the genetic makeup of the cultivars. Reports of the dissimilarity in the contents of sunflower oil have been attributed to genotypes, the type of analytical techniques used, or the type of species [17,22,24]. Fagbemi et al. [32] carried out a comparative study on the possible disparity in the essential oil contents of *Tamarindus indica* seeds from two analytical techniques (Soxhlet and hydro-distillation). The report established the presence of oil content and compositions in the former, thus corroborating the appropriateness of the adopted technique. It must be noted that the variation in the amount of oil found with each of the cultivars is within what is submitted for sunflower seeds in the literature to be between 36–50% [45]. In fact, a similar study conducted in Tunisia on 22 various sunflower accessions also reported a variation in the amount of oil obtained between 35.3 to 59.7% [17]. Besides, earlier studies have also established variations in the oil contents of the other parts of the plant, such as leaves and flowers (from Italy) [46] and stem and roots [47] from the same country. The composition of compounds within a plant may also differ due to the processing analysis [32]. The fatty acids obtained from respective cultivars are observed to be somehow similar but vary in their relative abundance. The differential may not be uncommon due to genetic and environmental factors or influences arising from the exposure of each cultivar to either a temperature or water regimen. The peak abundance of linoleic and oleic acids (polyunsaturated and monounsaturated fatty acids, respectively) among other fatty acids in most of the cultivars is welcoming; unsaturated fatty acids have been reported to majorly make up the largest components in sunflower seeds, accounting for 80–85% of the entire composition, while saturated fatty acids only take up 15–20% [48,49]. Additionally, the reports of Akayya [48] and Hosni et al. [17] from Egypt and Tunisia, respectively, earlier revealed similar findings, as established in this study, where the proportion of unsaturated was higher than the saturated while the abundance of linoleic and oleic acids (when combined) were reported to exceed 82 and 75%, respectively. Interestingly, the high abundance of linoleic acid in sunflower oil has been reported to be a consequence of its good quality and thus relates to its nutritional significance [50,51] and therapeutic benefits as it has the potential to reduce the risk of heart attack and cardiovascular diseases as well as in the lowering of the blood cholesterol levels [52].

Several factors, which may not be limited to the genotype of plants, access to soil nutrients, soil type and/ or agricultural practices, (geographical) location, and age, have been reported to influence the diversity and physiological trends of their microbial community structure [24]. However, an effort to determine whether or not the oil composition and/ or components of an oilseed plant (such as sunflower) contribute to its microbial diversity prompted the evaluation of fatty acids composition or abundance in relation to the microbial diversity of the samples as undertaken in this study. The lower abundance of important fatty acids (particularly linoleic acid) as observed in some cultivars may be suggested to be a consequence of the lower abundance of the studied microbial communities (particularly the fungi) at the genus level, hence, indicating a possible role of these fatty acids (linoleic and oleic acids) in causing lesser diversity in the fungi community structure.

The characterization of the endophytic microbiome from eco-samples is familiar and in recent times, explored in environmental research projects [53]. Despite the fact that the identification of many microbes still remains a mirage, yet to be cultured or identified, the arrival of next-generation sequencing and metagenomic methods have provided huge insights into microbial diversity compared to the culture-dependent approach, thus promoting a great deal of interest in phytobiome studies [54,55]. In fact, NGS has the advantage of identifying undetected or missed-out microorganisms in traditional culture methods [56]. The study identified dominant fungi such as Ascomycota, Basidiomycota, and Mucoromycota at the phyla level, though many ASVs were unidentified; this could partly be due to the unavailability of reference sequences in the used UNITE database [57] or possible novelty of the organisms. A report of severe underestimation and underrepresentation of fungal diversity has been previously established [58]. The presence of Ascomycota in all the cultivars may indicate their dominance in the community [29,59], suggesting it to be the core endophytic organism in the plant. Ascomycota is the largest community and most diverse group of fungi phyla with more than 93, 000 species [60]. Hence, the identification or the presence of this fungi in all the cultivars may not be surprising. The findings from this study were in tandem with the report of Rim et al. [61], who also identified Ascomycota (91.06%), Basidiomycota (5.95%), and Mucoromycota (2.97%) as the dominant fungi phyla from four Pinus species (*Pinus densiflora*, *Pinus koraiensis*, *Pinus rigida*, and *Pinus thunbergii*) from Korea. Additionally, Ascomycota was also largely represented or abundant (58.74%) in the roots of South Africa *Vigna unguiculata* (Cowpea) [62], followed by Basidiomycota. In reports [63,64] where fungi diversity of maize seed and sorghum cultivars (respectively) were studied, a higher abundance of Ascomycota in the two studies followed by Basidiomycota (in the maize study) was also established. The abundance of Ascomycota, particularly in selected cultivars from Agricol, Pannar, and LimaGrain may be indicative of their strongest capacity to build a community within the plant’s endosphere, arising from their ability to decompose plant biomass, maintenance of soil stability, carbon, nitrogen cycling (especially in arid regions) [65], and fermentation of foods [66], particularly in alcoholic beverages [67].

At the family level, while the 26 families representation in all the 20 cultivars with Pleosporaceae occurring and most abundant could be related to their possible plant growth and health-promoting effects [68,69], cultivars (AGSUN 8251) and (AGSUN 5102) are endowed with the most diverse fungi community composition, and this could be attributed to their possible superior nutritional profile, as a higher microbial diversity of endophytic samples was suggested to be influenced by nutritional composition among other factors such as the type of farming operation, differences in climatic or environmental factors, and organ differentiation [68]. Additionally, the lowest diversity observed with some cultivars may also be attributed to prevailing agricultural activities and geographical conditions [70]. While the relative abundance of Pleosporaceae had also been reported in *Cucurbita pepo* seeds (Pumpkin oil) cultivars from (Styria) Austria, variation (low and high) in the abundance of fungi communities among the cultivars at the family level was also established in this study [71]. The present study identified 17 genera from all the 20 cultivars where prominent genera such as *Alternaria*, *Cladosporium*, *Eremothecium*, *Penicillium*, *Aspergillus*, *Buckleyzyma,* and *Fusarium* are dominant in an appreciable amount in some of these cultivars in differential abundance; these genera were also found in an economically important crop, cowpea [62]. While these identified genera have been implicated as phytopathogens [62,72,73,74,75], interestingly, several species from some of these genera, particularly *Aspergillus*, *Penicillium,* and *Cladosporium,* have been identified as biocontrol agents. Typically, *Aspergillus flavus* in addition to other studied species according to the report of Boughalleb-M’Hamdi et al. [76] was established as a good biocontrol agent against the soil-borne fungi of melon and watermelon. In another report [77], *Aspergillus* species (*A. japonica*, *A. flavus*, *A. pseudoelegans* and *A. niger*) were also studied for their biocides’ activity on white mould disease of soybean caused by *Sclerotinia sclerotiorum*. Similarly, three *Cladosporium* species (*C. uredinicola*, *C. cladosporioides*, and *C. chlorocephalum*) were studied as potential biocontrol agents and were found to be aggressive on whiteflies (*Bemisia spp*) [78]. Islam et al. [74], in a study from Malaysia, corroborated the earlier Egyptian study when they reported the isolation of *Cladosporium cladosporiodes* from rice tested on the fungus *Bamesia tabaci* and found it to show excellent biocontrol properties. *Penicillium digitatum* was also found to be an effective biocontrol agent against soil fungi of melon and watermelon [76], *P. citrinum* was similarly submitted as a potential biocide for Sisal Bole Rot disease [79]. Additionally, *P. adametzoides* was found to control ochratoxin A, a fungus produced from grapes [80].

The earlier culture-based report on the identification of microbial fungi communities in sunflowers revealed genera such as *Alternaria*, *Fusarium*, *Aspergillus,* and *Penicillium* [81] which are also confirmed genera in this NGS study, buttressing other previous studies [35,62,82,83] where similar genera were identified in culture-dependent and culture-independent NGS methods on economically important crops such as cowpea, wheat, sugarcane, and rice. *Alternaria*, a cosmopolitan phytopathogen [62,84,85], is a ubiquitous fungus found in numerous habitats [84,85], possessing more than 300 species based on morphological and phylogenetic studies [84,86]. The fungus has been reported to produce numerous metabolites (more than 125), many of which are phytotoxic in nature and germane in the pathogenesis of plants [87], thus, contributing to survival threats to many foods and agricultural crops [88]. However, *Alternaria* is a fungus found everywhere, exhibiting rapid growth; it is dominant and easily colonizes or disperses any community [89]; its ubiquitous nature might explain why it was found in all the studied cultivars and its dominance, particularly in Agricol and Pannar cultivars which, however, could not be related to the observed variation in the oil compositions among the cultivars. Singh et al. [90], in a report could also not find a correlation between the severity of *Alternaria* blight and the fatty acid composition of mustard seed genotypes. However, reports of a high production of fatty acids especially linoleic acid have been attributed to fungi [87,91], it is therefore not surprising that the highest abundance of linoleic acids in virtually all the cultivars.

The presence of Proteobacteria in almost all the cultivars at the phyla level (and abundance in Pannar cultivars) was in line with the report of Fan et al. [92], where these bacteria were reported to be dominant in economically important crops such as soybean, maize, reed, alfalfa, miscanthus, and canarygrass. Interestingly, in a South African study, Proteobacteria was found to be dominant in the stem of a sunflower cultivar PAN 7160 CLP [29], indicating the similarity of their report with what was obtained in this present study, particularly with Pannar cultivars where complete domination by these bacteria was observed. Nevertheless, Proteobacteria, the gram-negative bacteria consisting of a body of spoilage, pathogenic, soil-borne, and human organisms [66], have been reported to be the largest bacteria community at the phyla level [93], also exemplifying its presence or abundance in all the cultivars. Firmicutes (a gram-positive bacteria associated with lactic acid bacteria), including Bacteroidetes, were also identified at the phyla level, with the former showing huge abundance (especially in Agricol cultivars). The identification of these groups of bacteria in this study was not surprising, as previous studies [29,94] on sunflower and other plants such as Glycyrrhiza species (*G. uralensis*, *G. inflata* and *G. glabra*), have similarly established their presence, attributed in the enhancement of growth and health sustainability of plants [68,95]. While there was limited identification of endophytic bacteria at the genus level in this study which was also maintained by Adeleke et al. [29], *Bacillus*, as represented by most of the cultivars in this study, was also included as one of the dominant genera including *Pseudomonas*, *Flavobacterium* in their study, indicating the similarity of the two studies. *Bacillus* is among the largest community of bacteria genera [96], comprising a large group of pathogenic and non-pathogenic bacteria with more than 200 species [97]. Most of its species have been reported to play a good role in agriculture and the environment; they are used as an alternative to getting rid of pathogens, thereby enhancing crop production. The application of *Bacillus* species in the promotion of plant growth and crop yields on a number of crops including tomato, wheat, pepper, maize, cucumber, soybean, common bean, and sunflower has been reported [98]. Nonetheless, the presence of *Bacillus* in this study may further suggest the possible exploration of its species as biocontrol agents [29]. Interestingly, *B. subtilis*, *B. megaterium*, *B. cereus*, *B. pumilus*, and *B. amyloliquefacians* among other *Bacillus* species have been explored as bioinoculants against citrus fruits diseases and several plant pathogens [69,83,99,100]. Additionally, species of other genera such as *Lactobacillus*, *Staphylococcus* etc., identified from cultivars in this study have also been explored as probable biocontrol agents in many scientific reports. Typically, *Lactiplantibacillus plantarum* was suggested as a possible biocontrol agent against phytopathogens (such as *Pectobacterium carotovorum*, *A. alternata*, *Fusarium oxysporum*, *A. solani*, *A. tenuissima*, *Colletotrichum coccode*, *Fusarium sambucinum*, *Streptomyces scabiei*, *Phoma exigua*, *Rhizoctonia solani*) of potato [101]. *L paracasei* (Lb38) and *L. brevis* (Lb99), as well as *L. plantarum* have also been found as useful bioinoculants in the food industry [102]. *Staphylococcus xylosus* was evaluated for its biocontrol effect against toxigenic moulds in meat [103].

## 5. Conclusions

The present study explored the use of a next-generation sequencing approach to identify the endophytic microbiome (fungi and bacteria) from 20 cultivars of South African sunflower seeds. While the study established no statistical differences in the oil composition of the cultivars, it therefore found differences existing in the microbial community structure, particularly in fungi with Agricol cultivars, especially AGSUN 5102 CLP exhibiting the most prominent diversity. AGSUN 5270 CLP has a high diversity level of bacteria community structure despite fewer or single bacteria representation in most of the cultivars at the genus level. The presence or abundance of unidentified organisms may be due to the novelty of the organisms and/ or lack of their sequence in the database used; hence further efforts towards possible detection and proper identification may be required, perhaps with better sequencing platforms such as PacBio. Further exploration of these species in biological control studies may come in handy towards the sustainability of agricultural systems. Therefore, it would be recommended that the study is replicated in three other Provinces (Free State, Gauteng, and Mpumalanga) where the seeds are largely produced in South Africa in an effort to have holistic information on the microbial community of the plant since geographical differences is a key factor that influences microbial diversity.

## Figures and Tables

**Figure 1 microorganisms-11-00988-f001:**
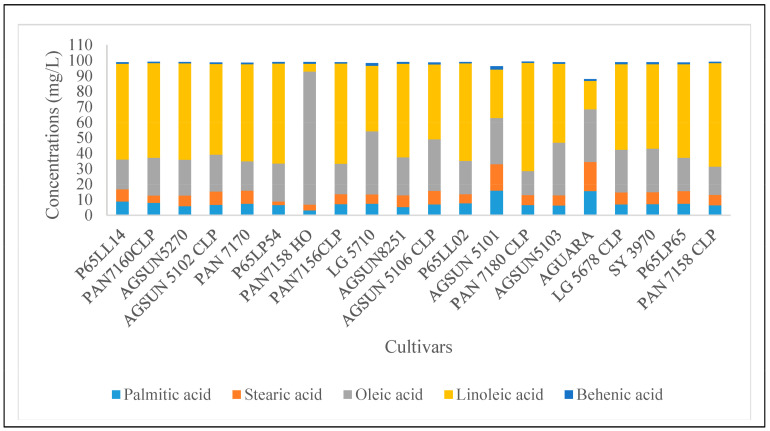
Fatty acid abundance in different cultivars of sunflower seeds oil. Linoleic acids dominated most of the cultivars except PAN7158 HO followed by oleic acid. AGUARA where the stearic acid was most abundant revealed a moderate proportion with other major fatty acids such as linoleic acids and palmitic acids. (Figure only depicted the five major fatty acids, different concentrations of other fatty acids are presented in Appendix A).

**Figure 2 microorganisms-11-00988-f002:**
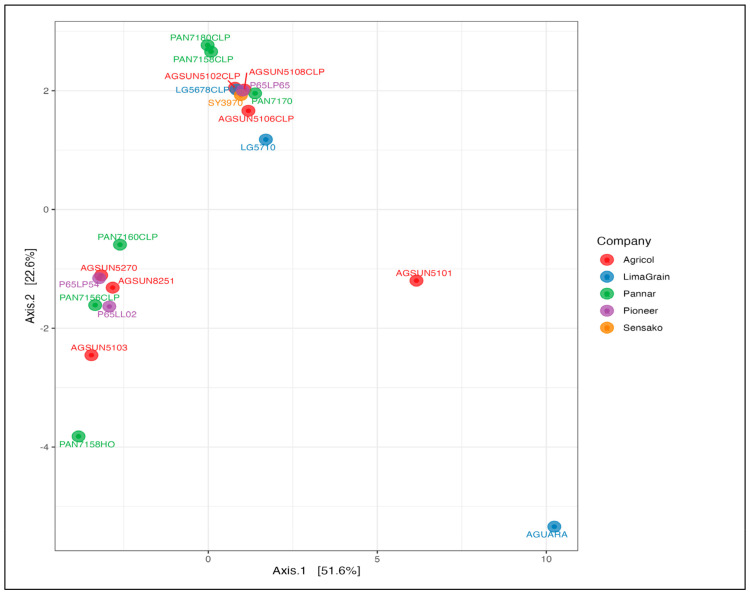
Principal component analysis of oil contents and fatty acids of sunflower seed cultivars obtained through gas-chromatography-mass spectrometry-fatty acid methyl esters (GCMS-FAMEs). A variation of 74.2% between the x and y axes (51.0% and 22.6%, respectively, existed with the percentage representing the difference in sample composition. The analysis revealed that the 20 seed cultivars clustered into two distinct orientations, though there was an outlier cultivar. The two clusters revealed representations from all the seed companies or brands. The cultivars in the first (above) group were closer together, indicating their close similarity in terms of oil and fat compositions while the cultivars in the second (below) group were slightly apart but above all, no significant relationship was observed among the cultivars.

**Figure 3 microorganisms-11-00988-f003:**
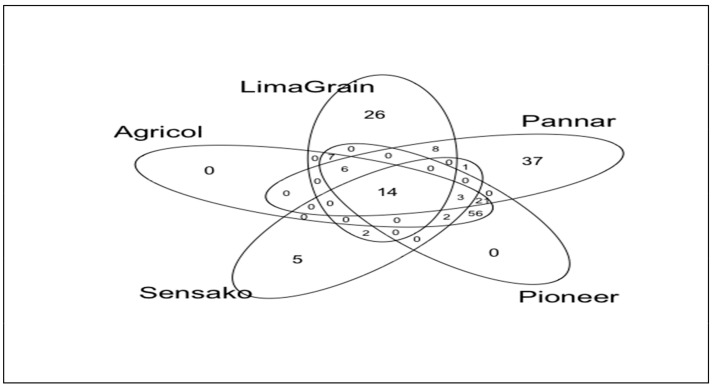
Venn diagram showing the shared and the unique amplicon sequence variants (ASVs) of fungi distribution among the various cultivar brands.

**Figure 4 microorganisms-11-00988-f004:**
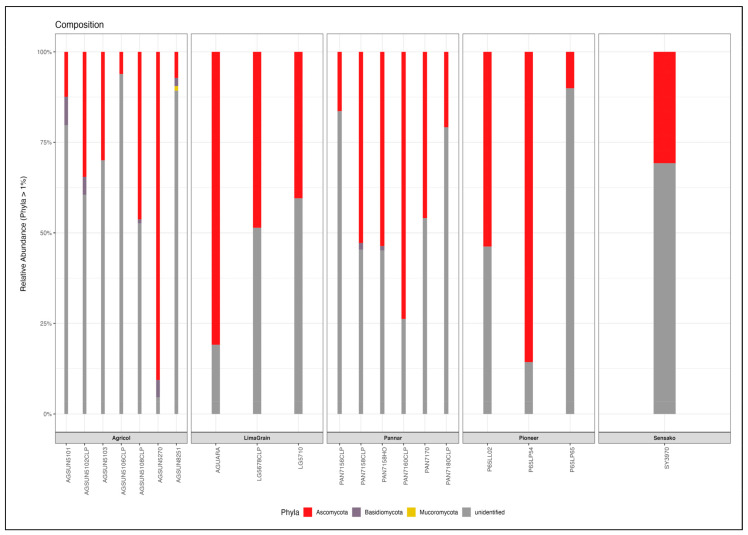
Relative abundance of the dominant fungi Phyla from different cultivars of South Africa *H. annus* seeds.

**Figure 5 microorganisms-11-00988-f005:**
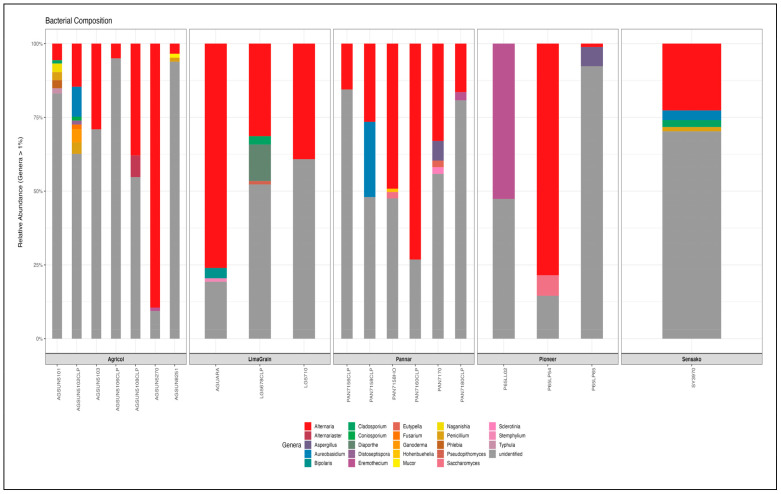
Relative abundance of the dominant fungi Genera from different cultivars of South Africa *H. annus* seeds. *Alternaria* was found in almost all the cultivars, though in varying fungi abundance in all the Agricol, Pannar, LimaGrain and Sensako. Cultivars such as AGSUN5102 CLP, LG5678 CLP, SY3970, PAN7158, AGSUN5101 and PAN7170 revealed some level of fungi diversity.

**Figure 6 microorganisms-11-00988-f006:**
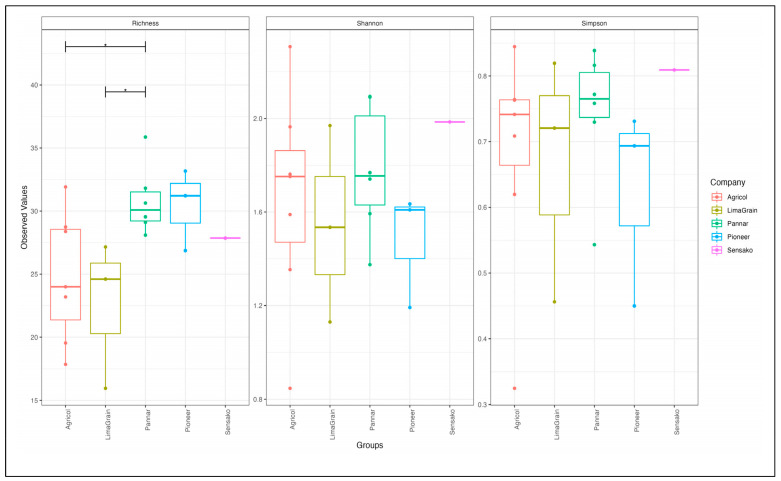
Fungi alpha diversity measurement for the observed number of ASVs (richness) for Chao1 (**left**), Shannon diversity (**middle**) and Simpson index (**right**) of sunflower seed cultivars.

**Figure 7 microorganisms-11-00988-f007:**
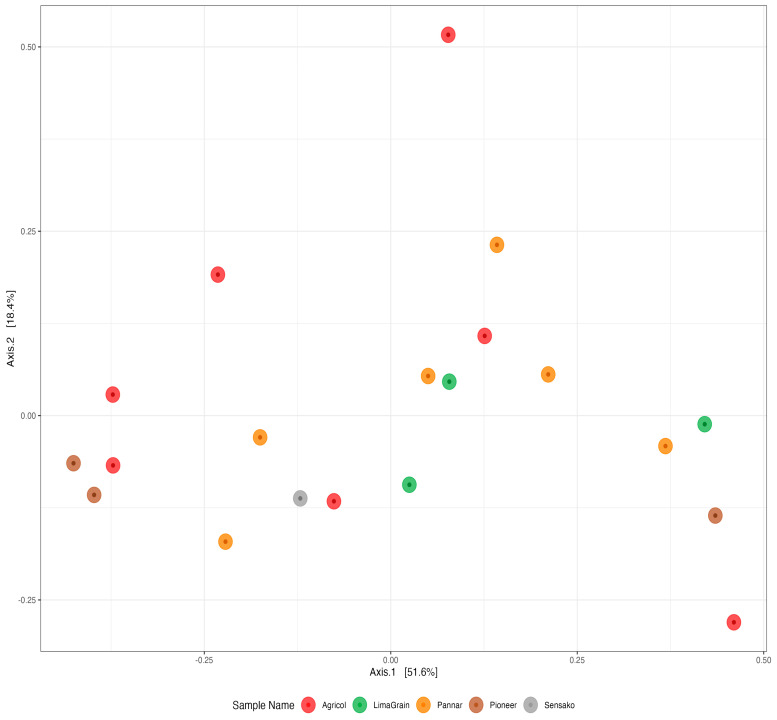
Principal co-ordinate analysis (PCoA) plot of beta diversity of fungi (based on Bray-Curtis dissimilarities) at the ASVs level of sunflower seeds. The PCoA data revealed that the first two PCoA components explained 18.4% and 51.6% of the variation, respectively.

**Figure 8 microorganisms-11-00988-f008:**
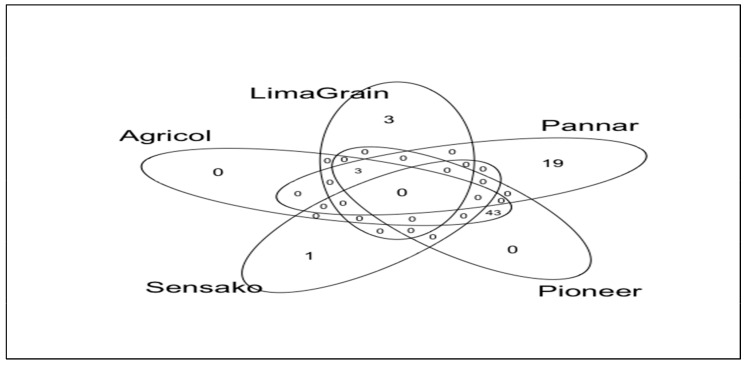
Venn diagram showing the shared and the unique ASVs of bacteria distribution among the various cultivar brands.

**Figure 9 microorganisms-11-00988-f009:**
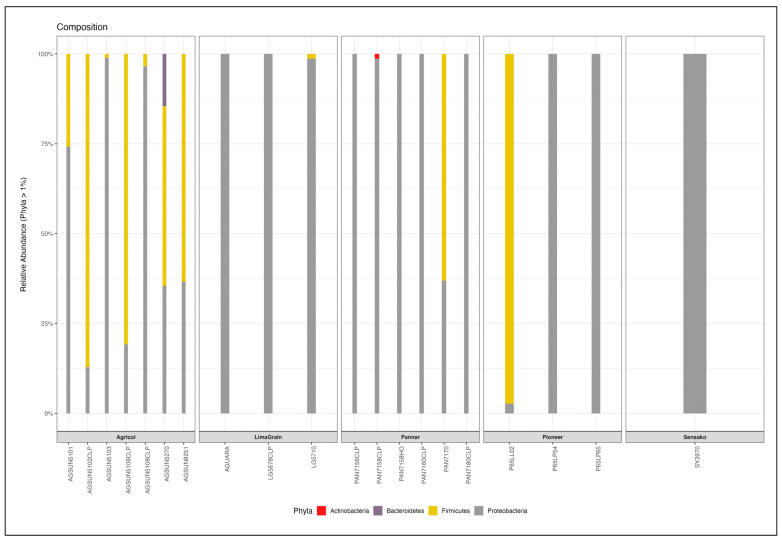
Relative abundance of the dominant bacteria Phyla from different cultivars of S. Africa *H. annus* seeds.

**Figure 10 microorganisms-11-00988-f010:**
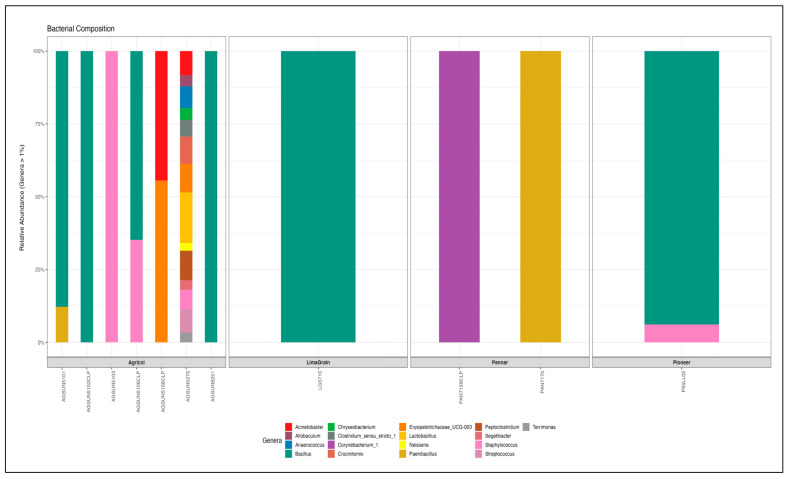
Relative abundance of the dominant bacteria Genera from different cultivars of South Africa *H. annus* seeds. *Bacillus* appeared in 7 of the cultivars, with the highest abundance in most. Interestingly, AGSUN5270 was the only cultivar with the highest degree of bacteria diversity, with 14 genera of bacteria community.

**Figure 11 microorganisms-11-00988-f011:**
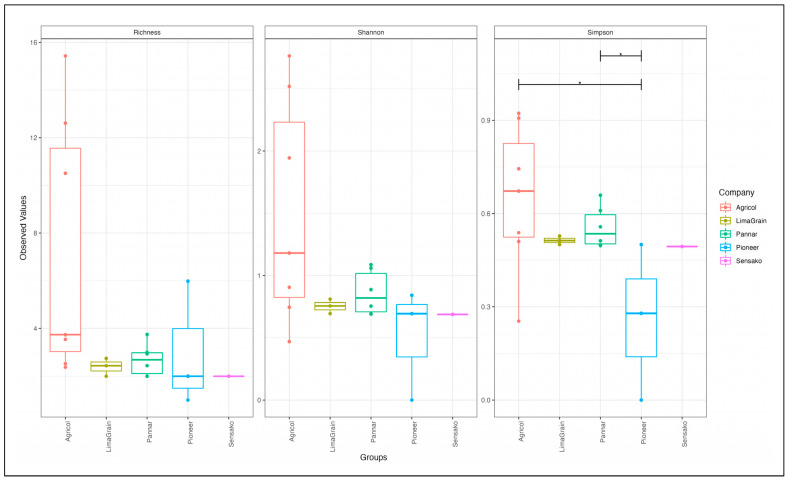
Bacteria alpha diversity measurement (observed number of ASVs (richness) for Chao1 (**left**), Shannon diversity (**middle**) and Simpson index (**right**) of sunflower seed cultivars.

**Figure 12 microorganisms-11-00988-f012:**
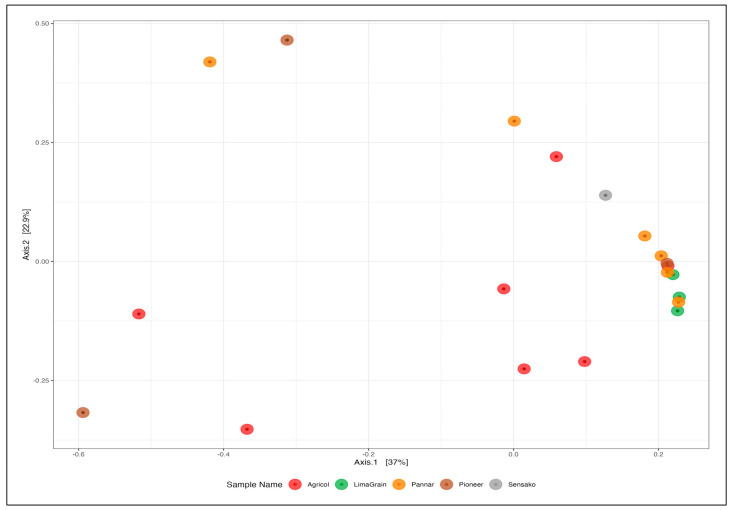
PCoA plot of beta diversity of bacteria (based on Bray-Curtis dissimilarities) at the ASVs level of sunflower seeds. The PCoA data revealed that the first two PCoA components explained 22.9% and 37% of the variation, respectively.

**Table 1 microorganisms-11-00988-t001:** Percentage oil composition of sunflower seed cultivars.

Brands	Assign. No	Cultivar Names	Oil Contents (%)
AGRICOL	4	AGSUN5270	52.8
	5	AGSUN 5102 CLP	45.0
	11	AGSUN8251	44.9
	12	AGSUN 5106 CLP	50.0
	14	AGSUN 5101	40.8
	16	AGSUN5103	48.0
PANNAR	3	PAN7160CLP	51.7
	6	PAN 7170	52.8
	8	PAN7158 HO	42.4
	9	PAN7156CLP	52.0
	15	PAN 7180 CLP	46.9
	21	PAN 7158 CLP	42.4
PIONEER	2	P65LL14	47.0
	7	P65LP54	43.0
	13	P65LL02	42.9
	20	P65LP65	41.0
LIMAGRAIN	10	LG 5710	50.5
	17	AGUARA	44.0
	18	LG 5678 CLP	47.9
SENSAKO	19	SY 3970	41.8

## Data Availability

Additional data for this study can be found in the Appendix A section.

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
