# Peer review of "Chemical Characterization and Metagenomic Identification of Endophytic Microbiome from South African Sunflower (Helianthus annus) Seeds"

_microorganisms, 2023, doi:10.3390/microorganisms11040988_

Round 1
Reviewer 1 Report
The manuscript ID microorganisms-2300849 entitled " Chemical characterization and metagenomic identification of endophytic microbiome from South African sunflower (Helianthus annus) seeds", is an interesting study. But my suggest it needs to a major revision to be published in this journal. I do have some comments about the manuscript and data interpretation/discussion that could improve the overall quality of the manuscript:
1- The authors repeated the term biological control more than once section, such as the abstract, the introduction, the discussion, and the conclusion, although this study did not address this part at all. I would only like to ask the authors why this term was referred to in this manuscript?
2- I suggest writing a special section for all the abbreviations because they are too many in the manuscript.
3- Abstract : biological control and plant pathogens must be deleted from this section.
4- Introduction : With the length of the introduction and the large number of references (27), the authors neglected many points, including the importance of endophytes to the plant and its importance in general.
Lines 64:66: revise and check this rank of plant.
5- Materials and methods: Wonderfully written. But there is quarry (line 99) why the authors treated the seeds with fungicides ? and what is the type of fungicide used?
6- Lines 152:154: the seed coat removed and subjected to 70% ethanol for 1 minute ? ( why)???????????
7- Results : Line 195 : (41-53%) revise and check , in table ( 41-52.8%).
8- Line 255 : what is the ( ASVs) ?
9- Figures are complex and difficult to understand and require an explanatory explanation. However, in this way, they need a statistician to understand them.
10- Fungal and bacterial diversity need to In-depth explanation.
11- Discussion : lines 495and 496 It is better to put it in the introduction section.
12- Conclusion section : Why was biological control explained in this section when there were no results or methods related to it in this study?
Author Response
Dear Reviewer,
We appreciated your effort in reviewing our paper and the valuable comments presented to us to address. Please in line with the aforementioned, kindly find the attached document as our responses to the raised concerns. We hope they will meet your desired satisfaction. Thank you

Reviewer 2 Report
Dear authors, the topic presented in the manuscript "Chemical characterization and metagenomic identification of endophytic microbiome from South African sunflower (Helianthus annus) seeds" is important in the current context of proposal new biocontrol techniques and the correlation between plant composition and its microbiome.
There are some changes that will improve the current form of the manuscript and will increase the readability of the results.
Introduction offers multiple information that sustain the manuscript.
The lines 89-95 need to be presented as a separate paragraph. The aim of the manuscript should be clearly stated. The authors can use a sentence like: "The research aimed to analyze in detail the seed microbiome structure in 20 South African sunflower cultivars." After that, present the objectives, each in a separate sentence: Chemical composition. Microbiome structure - fungi and bacteria. The correlation between those two, maybe as another objective. It can be added a sentence to state that the study is done on seeds, after the removal of coats, which will sustain the idea of a future interaction of this microflora with the soil microflora.
Material and Methods describe in detail the procedures used by the authors.
Results section
3.1. Fats composition - chemical composition should be presented with ANOVA and a post-hoc test, means±s.e. (or s.d.). Use the average of replications for each determination. This section should be expanded. Vectors of each type of fatty acid can be projected on PCA and will help the interpretation.
All subsections from results should be expanded, in this form there are large figures but with a reduced interpretation. Add a multiple comparison test for Figure 6 and 11 if possible.
Discussion section connect the results with the international literature and it has an appropriate length.
Conclusion section - this section should present in separate sentences the main findings of the research, with values and species names.
Overall, the manuscript presents and interesting research and results.
Author Response

(The authors gave the same response as above.)

Round 2
Reviewer 1 Report
The authors made some required modifications to improve the manuscript.
Reviewer 2 Report
Dear authors, the new form of the manuscript present your findings in an improved form.